# Influence of Acoustic Parameters and Sonication Schemes on Transcranial Blood–Brain Barrier Disruption Induced by Pulsed Weakly Focused Ultrasound

**DOI:** 10.3390/pharmaceutics14061207

**Published:** 2022-06-06

**Authors:** Yu-Hone Hsu, Wei-Chung Lee, Shing-Shung Chu, Meng-En Chao, Kuo-Sheng Wu, Ren-Shyan Liu, Tai-Tong Wong

**Affiliations:** 1Division of Neurosurgery, Kaohsiung Veterans General Hospital, Zuoying, Kaohsiung 813, Taiwan; hsuyhmail@gmail.com; 2School of Nursing, National Taipei University of Nursing and Health Sciences, Taipei 112, Taiwan; 3Graduate Institute of Clinical Medicine, College of Medicine, Taipei Medical University, Taipei 110, Taiwan; b90606043@gmail.com (W.-C.L.); dog52037@gmail.com (S.-S.C.); chaomengen@gmail.com (M.-E.C.); abel1063@gmail.com (K.-S.W.); 4Department of Biomedical Imaging and Radiological Sciences, National Yang Ming Chiao Tung University, Taipei 112, Taiwan; 5PET Center, Department of Nuclear Medicine, Taipei Veterans General Hospital, Taipei 112, Taiwan; 6Department of Nuclear Medicine, Cheng Hsin General Hospital, Taipei 112, Taiwan; 7Pediatric Brain Tumor Program, Taipei Cancer Center, Taipei Medical University, Taipei 110, Taiwan; 8Division of Pediatric Neurosurgery, Department of Neurosurgery, Taipei Medical University Hospital, Taipei Medical University, Taipei 110, Taiwan; 9Neuroscience Research Center, Taipei Medical University Hospital, Taipei 110, Taiwan

**Keywords:** blood–brain barrier, brain drug delivery, brain tumors, neurodegenerative diseases, Evans blue dye, focused ultrasound, pulsed-wave low-dose ultrasound, microbubbles

## Abstract

Pulsed ultrasound combined with microbubbles use can disrupt the blood–brain barrier (BBB) temporarily; this technique opens a temporal window to deliver large therapeutic molecules into brain tissue. There are published studies to discuss the efficacy and safety of the different ultrasound parameters, microbubble dosages and sizes, and sonication schemes on BBB disruption, but optimal the paradigm is still under investigation. Our study is aimed to investigate how different sonication parameters, time, and microbubble dose can affect BBB disruption, the dynamics of BBB disruption, and the efficacy of different sonication schemes on BBB disruption. **Method:** We used pulsed weakly focused ultrasound to open the BBB of C57/B6 mice. Evans blue dye (EBD) was used to determine the degree of BBB disruption. With a given acoustic pressure of 0.56 MPa and pulse repetitive frequency of 1 Hz, burst lengths of 10 ms to 50 ms, microbubbles of 100 μL/kg to 300 μL/kg, and sonication times of 60 s to 150 s were used to open the BBB for parameter study. Brain EBD accumulation was measured at 1, 4, and 24 h after sonication for the time–response relationship study; EBD of 100 mg/kg to 200 mg/kg was administered for the dose–response relationship study; EBD injection 0 to 6 h after sonication was performed for the BBB disruption dynamic study; brain EBD accumulation induced by one sonication and two sonications was investigated to study the effectiveness on BBB disruption; and a histology study was performed for brain tissue damage evaluation. **Results:** Pulsed weakly focused ultrasound opens the BBB extensively. Longer burst lengths and a larger microbubble dose result in a higher degree of BBB disruption; a sonication time longer than 60 s did not increase BBB disruption; brain EBD accumulation peaks 1 h after sonication and remains 81% of the peak level 24 h after sonication; the EBD dose administered correlates with brain EBD accumulation; BBB disruption decreases as time goes on after sonication and lasts for 6 h at least; and brain EBD accumulation induced by two sonication increases 74.8% of that induced by one sonication. There was limited adverse effects associated with sonication, including petechial hemorrhages and mild neuronal degeneration. **Conclusions:** BBB can be opened extensively and reversibly by pulsed weakly focused ultrasound with limited brain tissue damage. Since EBD combines with albumin in plasma to form a conjugate of 83 kDa, these results may simulate ultrasound-induced brain delivery of therapeutic molecules of this size scale. The result of our study may contribute to finding the optimal paradigm of focused ultrasound-induced BBB disruption.

## 1. Introduction

The blood–brain barrier (BBB) consists of tight junctions between endothelial cells of brain capillaries, the basement membrane, and foot processes of astrocytes. This three-layered structure forms a barrier to protect the brain from circulating toxic molecules. Normally only lipophilic molecules smaller than 500 Da can pass through the BBB [1,2,3,4,5,6,7,8,9,10]. Although this barrier is efficient for brain protection, it limits therapeutic molecules to enter the brain for disease treatment. For example, Herceptin (trastuzumab), a 145 kDa monoclonal antibody, has been successfully used in breast cancer treatment, but its effectiveness is limited in the case of brain metastasis because its entry to the brain is blocked by the BBB, and when Herceptin is administered systemically, its concentration in cerebrospinal fluid is only 0.3% of that in plasma [11].

Strategies to overcome the BBB includes intraventricular injection, convection enhanced delivery, carotid artery injection of hyperosmolar solution, and intranasal delivery to cerebrospinal fluid. These approaches are too invasive or results in only limited drug diffusion [6,9,10,12,13]. Some strategies modify drugs by drug lipidization, protein cationization, or chimeric peptide technology to enhance their BBB permeability, but therapeutic applications of these agents are under limited development [14]. Blood–brain barrier disruption (BBBD) induced by carotid artery injection of hyperosmolar solution has been used in human clinical trials for chemotherapy delivery to the brain to treat central nervous system (CNS) lymphoma, CNS germ cell tumors, cancer brain metastasis, and gliomas; the results were inspirational but the procedure is invasive and complications cannot be ignored, including arterial dissection, pulmonary emboli, renal toxicity, stroke, seizure, myelosuppression, and hearing loss [15,16].

Pulsed ultrasound combined with microbubbles (MBs) has been proven to open the BBB noninvasively and temporarily with limited brain tissue damage [1,17,18,19,20,21,22]. The mechanism of BBBD induced by pulsed ultrasound combined with MBs includes an acoustic radiation force, microstreaming formation, stable cavitation, and inertial cavitation. The radiation force pushes MBs to hit the capillary wall and then distract the tight junctions between endothelial cells. The MB oscillates in the acoustic field and induces a microstreaming formation nearby it, which exerts a shearing force on the capillary wall and then distracts the tight junctions between the endothelial cells. The stable cavitation refers to oscillation of the MB volume; when a MB expands large enough, it distracts the tight junctions. The inertial cavitation refers to a several-fold expansion of the MB volume followed by sudden collapse; this activity emits shock waves that results in capillary wall damage and plasma content extravasations [2,5,7,8].

Minimizing the variables improves safety during BBB disruption induced by FUS [23]. With an appropriate acoustic pressure setting, 2000 kDa dextran molecules can be delivered to the brain in an animal model [24]. Large molecular weight therapeutic agents that have been delivered to the brain successfully in animal models using this technique includes anti-dopamine D4 receptor antibodies, Herceptin, liposome-encapsulated doxorubicin, genetically engineered viral gene vector, anti-amyloid beta antibodies, and some chemotherapy agents [25]. Most of the associated studies used focused ultrasound to open the BBB [26]. The advantage of focused ultrasound is to open the BBB in a small selective area; it is suitable for brain tumor treatments in which drug delivery should be focal and targeted, but for disease such as infiltrative brain tumors, multiple brain metastasis, or lysosomal storage disease with CNS involvement, the therapeutic agents should be delivered to the brain diffusely [27]; thus, an ultrasound device to open the BBB broadly will be more suitable.

The efficacy of pulsed ultrasound on BBBD depends on the acoustic parameter setting, including the acoustic pressure, pulse repetitive frequency (PRF), burst length, MB dose, and sonication time.

The objective of this study was to use pulsed weakly focused ultrasound to open the BBB extensively, and investigate how different ultrasound parameters (burst length, MB dose, and sonication time) influence the BBBD, time–response relationship, and dose–response relationship of the delivered agent after BBBD, as well as the dynamics of BBBD and efficacy of the different sonication schemes on BBBD.

## 2. Materials and Methods

### 2.1. Animal Preparation

The animal work, including housing, caring, and experimental procedures, were carried out in strict accordance with the recommendations in the Guide for the Care and Use of Laboratory Animals of the National Institutes of Health (Bethesda, MD, USA). The experimental protocols were approved by the Institutional Animal Care and Use Committee of Cheng Hsin General Hospital. The C57BL/6 mice (B6 mice) were obtained from the National Laboratory Animal Center; the mice were treated when they had body weight of 19–25 g. All surgery was performed under adequate anesthesia, and all efforts were made to minimize suffering. The study was conducted from 1 August 2016 to 31 July 2019.

### 2.2. Ultrasound Equipment

We used a 1 MHz single-element transducer (A392S, Panametrics, Waltham, MA, USA, diameter 38 mm, radius of curvature 63.5 mm) to generate the ultrasonic wave. The transducer was mounted to a cone that was filled with distilled and degassed water and capped by a polyurethane membrane. We fixed the transducer–cone apparatus to a stereotactic instrument (David Kopf Instruments, Tujunga, CA, USA), which allowed the cone tip to move to aim at different sonication sites. We connected the transducer to a function generator (33220A, Agilent Technologies, Santa Clara, CO, USA) and a power amplifier (75A250A, Amplifier Research, Souderton, PA, USA) to drive it. The mouse was lying prone with its head beneath the cone tip. We smeared the mouse head with ultrasound transmission gel to ensure ultrasound energy transmission (Figure 1A).

### 2.3. Study Arrangement

The study was divided into three parts. Part one was to investigate the effectiveness of different ultrasound parameters on BBBD as well as the time–response and dose–response relationships of brain drug delivery after BBBD (Table 1); part two was to investigate the dynamics of BBBD; and part three was to compare the effectiveness of one-spot and two-spot sonication on BBBD. The degree of BBBD was determined by Evans blue dye (EBD) delivered to the brain. In part one of this study, the B6 mice were divided into five groups (Table 1). Then, with a given acoustic pressure of 0.56 MPa and pulse repetitive frequency (PRF) of 1 Hz, we investigated the effectiveness of burst lengths of 10, 30, and 50 ms, a microbubble dose of 100, 150, 200, and 300 μL/kg, and sonication time of 60, 90, 120, and 150 s on BBBD; time–response relationship of brain EBD delivery of 1, 4, and 24 h after sonication; and dose–response relationship of brain EBD delivery of EBD 100, 150, and 200 mg/kg injection after BBBD (Figure 2A). In part two of this study, with an acoustic parameter of 0.56 MPa, PRF of 1 Hz, burst length of 50 ms, MB dose of 300 μL/kg, and sonication time 60 s, EBD 100 mg/kg were injected at different time intervals (0, 0.5, 1, 2, 3, 4, and 6 h) after sonication to investigate the dynamics of BBBD (Figure 2B); the mice were euthanized one hour after EBD injection. In part three of this study, we compared the efficiency of one-spot sonication and two-spot sonication on BBBD. The mice were divided into 3 groups: Group 1 received EBD 100 mg/kg plus an MB 150 μL/kg injection only; Group 2 and Group 3 received EBD 100 mg/kg plus an MB 150 μL/kg injection followed by one-spot sonication and two-spot sonication, respectively, where each spot was given with an acoustic pressure of 0.56 MPa, PRF of 1 Hz, burst length of 10 ms, and sonication time of 60 s (Figure 2C,D); the mice were euthanized four hours after sonication.

### 2.4. Animal Procedures

We anesthetized the mice by Zoletil 20 mg/kg (Zoletil^®^50, Virbac Laboratories, Carros, France) plus Xylazine 5 mg/kg (Rompun™, Bayer, Shawnee Mission, KS, USA) intraperitoneal injection, and cannulated tail veins for EBD and MBs injection. We shaved the heads of mice and fixed them on the stereotactic apparatus. For one sonication (in Parts 1, 2, and 3 of this study), we put the cone tip on the point 2 mm left lateral to the bregma (Figure 1B); for two sonications, we put the cone tip on the first point, which was 1 mm posterior to the one-spot sonication point, and then moved to the second point, which was 1 mm anterior to the one-spot sonication point (Figure 1C). After the mouse and the cone were in position, we injected a mixture solution of MBs (SonoVue^®^, Bracco, Amsterdam, The Netherlands) and EBD (Sigma-Aldrich, St. Louis, MO, USA) through the tail vein, and then sonicated immediately (Figure 2). For mice treated with two-spot sonication, we injected the mixture solution of MB 150 μL/kg and EBD 100 mg/kg in two half-injections; each injection was followed by a sonication (Figure 2D), so that the total dose of EBD and MBs used for the two sonications was the same as that used in one sonication. After euthanization, we perfused the mice with normal saline through a heart cannulation and then harvested the brains for slicing and EBD quantification.

### 2.5. EBD Quantification

We weighed mice brain samples and placed them in trichloroaceticacid solution, and then homogenized and centrifuged them to extract EBD. The extracted EBD was diluted with 95% ethanol (1:3) and then placed in a multiwell-plate-reading fluorometer (X-Zell Biotec, Bangkok, Thailand). We measured fluorescence at 680 nm with excitation at 620 nm and converted the reading value to concentration by a standard curve derived from preformed concentrations of EBD.

### 2.6. Statistical Analysis

We used SPSS software for data analysis and presented the data as the mean ± standard error of the mean. We used Tukey tests for comparisons between paired samples and one-way ANOVA for comparisons between three or more samples. *p* < 0.05 was considered to be significant.

### 2.7. Histology Study

Eleven mice treated with one-spot or two-spot sonication were subjected to a brain histology study for tissue damage evaluation. The ultrasound parameters, sonication procedures, and MB dose used were the same as shown in Figure 2C,D. The mice were euthanized and perfused with normal saline 4 h after sonication. The brain was removed and cut into four coronal slices, fixed in 10% neutral formalin, embedded in paraffin, sectioned into slices of 4 μm thickness, and then stained with hematoxylin and eosin. The slices were examined under a light microscope. Brain tissue damage was graded following the criteria proposed by Hynynen et al. [28]: the grade ranges from 0 to 3, where “0” corresponds to no detected damage; “1” corresponds to one to a few tiny red blood cell extravasations; “2” corresponds to petechial hemorrhages or mild damage to the brain parenchyma; and “3” corresponds to hemorrhagic or nonhemorrhagic local lesions.

## 3. Results

### 3.1. Ultrasound Parameters, Time–Response Relationship, and Dose–Response Relationship

#### 3.1.1. The Degree of BBBD Was Determined by Brain EBD Accumulation

In part one of this study, the effectiveness of different experimental conditions on BBBD was investigated (Table 1).

The results showed that (1) BBBD increased as burst length increased. With acoustic pressure of 0.56 MPa, PRF of 1 Hz, sonication time of 60 s, MB of 150 μL/kg, and EBD of 100 mg/kg injected, burst lengths of 10 ms, 30 ms, and 50 ms were used for BBBD (Figure 2A). Brain EBD accumulation increased as burst length increased from 10 ms to 50 ms (Figure 3A); (2) A large MB dose injected resulted in broad BBBD. With acoustic pressure of 0.56 MPa, PRF of 1 Hz, sonication time of 60 s, burst length of 10 ms, and EBD of 100 mg/kg injected, MB doses of 0, 100, 150, 200, 300 μL/kg were used for BBBD (Figure 2A). Brain EBD accumulation was similar with MB doses 100, 150, and 200 μL/kg, and increased significantly with MB dose 300 μL/kg (Figure 3B); (3) To increase the sonication time did not necessarily increase BBBD. With acoustic pressure of 0.56 MPa, PRF of 1 Hz, burst length of 10 ms, MB of 150 μL/kg, and EBD of 100 mg/kg injected, sonication times of 60, 90, 120, and 150 s were used for BBBD (Figure 2A). There was no significant difference in brain EBD accumulation as the sonication time increased from 60 s to 150 s (Figure 3C); (4) Brain EBD accumulation remained at a high level 24 h after delivery. With acoustic pressure of 0.56 MPa, PRF of 1 Hz, burst length of 10 ms, sonication time of 60 s, MB of 150 μL/kg, and EBD of 100 mg/kg injected, brain EBD accumulation was measured at 0, 1, 4, and 24 h after sonication (Figure 2A); it peaked one hour after sonication and persisted for additional three hours, and remained at 81% of the peak level 24 h after delivery (Figure 3D); (5) Brain EBD accumulation was dependent on the EBD dose injected. With acoustic pressure of 0.56 MPa, PRF of 1 Hz, burst length of 10 ms, sonication time of 60 s, and MB of 150 μL/kg injected, EBD doses of 100, 150, and 200 mg/kg were injected (Figure 2A). Brain EBD accumulation increased as the EBD injection dose increased (Figure 3E), and there was a correlation between them with a correlation coefficient of 0.7836 (Figure 3F). The region of BBBD was blue stained by EBD (Figure 4); the characteristics of the BBBD area distribution induced by weakly focused ultrasound are broad and non-focused, involving the whole thickness of the brain in the acoustic field.

#### 3.1.2. BBBD Dynamics

In part two study of this, we investigated the dynamics of pulsed ultrasound-induced BBBD. With acoustic pressure 0.56 of MPa, PRF of 1 Hz, burst length of 50 ms, sonication time of 60 s, and MB of 300 μL/kg injected (Figure 2B), EBD 100 mg/kg was injected at 0, 0.5, 1, 2, 3, 4, and 6 h after sonication; the mice were then euthanized 1 h after EBD injection. The degree of BBBD was the largest immediately after sonication, then decreased and remained stable at 1 to 4 h after sonication. At 6 h after sonication, BBBD was still significantly higher than that of the control group, suggesting that BBBD can last for 6 h at least (Figure 5).

#### 3.1.3. Efficiency of Different Sonication Schemes

In part three study we compared the efficiency of one-spot sonication and two-spot sonication on BBBD (Figure 2C,D). The ultrasound parameters used for each spot was acoustic pressure of 0.56 MPa, RPF of 1 Hz, burst length of 10 ms, and sonication time of 60 s. The total dose of EBD and MBs used in one-spot and two-spot sonication treatment were the same. As shown in Figure 3, two slices were significantly stained in the one-spot sonication scheme, and all four slices were significantly stained in the two-spot sonication scheme. Quantitatively, two-spot sonication resulted in a 74.8% increase in brain EBD accumulation than one-spot sonication. EBD delivered to the brain on the ultrasound treated side was 2.80-fold and 3.87-fold that on the untreated side after one-spot and two-spot sonication treatment, respectively (Figure 6).

Group 1 (*n* = 9): EBD 100 mg/kg plus MB 150 μL/kg injection only, no sonication. Group 2 (*n* = 15): One-spot sonication. MB 150 μL/kg plus EBD 100 mg/kg was injected in one injection. Group 3 (*n* = 6): Two-spot sonication. MB 150 μL/kg plus EBD 100 mg/kg was injected in two half-injections, each injection was followed by sonication.

#### 3.1.4. Histology Study

Eight mice treated with one-spot sonication and three mice treated with two-spot sonication were subjected to brain histology study for tissue damage evaluation. The results are listed in Table 2.

## 4. Discussion

BBBD was confirmed by EBD extravasations to the brain tissue in our study. After being injected to systemic circulation, EBD (961 Da) binds to plasma albumin (69 kDa) preferentially and completely to form a high molecular weight conjugate of 71–83 kDa (8–14 EBD molecules bind to one plasma albumin molecule) [29] and cannot pass the BBB under normal circumstance [30].

In contrast to strongly focused ultrasound, the characteristics of BBBD area distribution induced by weakly focused ultrasound are broad and non-focused, involving the whole thickness of the brain in the acoustic field (Figure 3). For diseases such as focal brain tumors, the therapeutic agents should be delivered to a small selective area. Using strongly focused ultrasound to open the BBB of a small targeted area is appropriate, but for diseases such as infiltrative brain tumors, multiple brain metastasis, or lysosomal storage disorders with CNS involvement (in which all brain cells lack a specific enzyme), extensive delivery of therapeutic agents to the brain is needed [27]; using weakly focused ultrasound to open BBB broadly will be more suitable.

In part one of this study, we investigated the effectiveness of different ultrasound parameters on BBBD, time–response relationship, and dose–response relationship of brain EBD delivery. With a given acoustic pressure of 0.56 MPa and PRF of 1 Hz used, we found that (1) a longer burst length induces more brain EBD accumulation (Figure 3A). Previous studies reveal that with a MB dose of 50 μL/kg and burst length of 10 ms, longer burst lengths increase the magnitude of BBBD, but no additional effect found beyond this point [31,32,33]. Our study shows that with an MB dose of 150 μL/kg, BBBD increased as burst length increased from 10 ms to 50 ms. We speculate that increased burst length will not increase the degree of BBBD if each MB reaches its maximal activity in the acoustic field; therefore, with an adequate MB dose, a longer burst length will induce a higher degree of BBBD. (2) BBBD is similar with an MB dose of 100–200 μL/kg and increased with an MB dose of 300 μL/kg (Figure 3B). The pulsed ultrasound cannot open the BBB without aid of MBs. With an MB dose 300 μL/kg, brain EBD accumulation was 1.95-fold that of MB dose 150 μL/kg. Since BBBD is achieved by MB activities within the capillaries in the acoustic field, such as stable cavitation or microstreaming formation, more MBs injected will cause more acoustic activities exerted on the capillary wall, therefore resulting in a higher degree of BBBD. Yang et al. and Treat et al. have demonstrated that by use of pulsed ultrasound, larger doses of MBs induced a higher degree of BBBD [21,34]; our findings are consistent with their results. (3) To increase the sonication time over 60 s did not increase BBBD (Figure 3C). The MBs were eliminated 40–50% within one minute after injection; the elimination is independent of dose [35]. Such a rapid decay of MBs explains why a longer sonication time did not necessarily induce a higher degree of BBBD. (4) Brain EBD accumulation reached a plateau one hour after sonication and persisted for an additional 3 h; the delivered EBD still retained 81% of the peak level 24 h after sonication (Figure 3D). In contrast, the half-life of EBD in plasma is only 2–6.5 h [30,36]. We speculate that this difference in EBD–albumin conjugate clearance time is due to the reversibility of BBBD. The delivered EBD–albumin conjugate is confined to the brain after BBB closure and therefore cannot be cleared through the usual way in the circulation. This finding suggests that some therapeutic agents delivered to the brain by ultrasound may have a longer acting time than they were in plasma. (5) Brain EBD accumulation increased as EBD dose injected increased (Figure 3E), and there was a good correlation between them with a correlation coefficient of 0.7836 (Figure 3F). The brain EBD accumulation induced by EBD 200 mg/kg injection was 6.17-fold that induced by EBD 100 mg/kg injection. Among the parameters discussed above, the EBD dose injected was the most influential factor if the BBB was opened. Previous studies show that brain delivery of Herceptin and doxorubicin induced by ultrasound correlates to MR signal intensity change and can be predicted by it [21,37]; our study shows that it is possible to predict brain drug delivery over a certain range simply by drug dose administered in the case of BBBD.

In part two of this study, we investigated the dynamics of BBBD. With an acoustic pressure 0.56 of MPa, PRF of 1 Hz, burst length of 50 ms, sonication time of 60 s, and MB dose of 300 μL/kg, EBD 100 mg/kg was injected 0, 0.5, 1, 2, 3, 4, and 6 h after sonication. The mice were euthanized 1 h after EBD injection. BBBD was the largest immediately after sonication, then decreased and remained stable at 1 h to 4 h, and then decreased again. At 6 h after sonication, brain EBD delivery was still significant (Figure 5), suggesting that pulsed ultrasound induced BBBD can last for 6 h at least. This duration may serve as a time window for brain delivery of therapeutic agents. Previous studies investigating the duration of BBBD have demonstrated that BBBD can last for 2 h in a pig model [38], 4 h in a rat model [39], and 6 h in a rabbit model [2,40]. Our findings are similar to these studies.

In part three of this study, we investigated the efficiency of one-spot sonication and two-spot sonication schemes on BBBD (Figure 2C,D). As Figure 3 shows, with the same total dose of MBs and same ultrasound parameters used, the BBBD area induced by two-spot sonication involved more brain slices than that induced by one-spot sonication. Quantitatively, the brain EBD accumulation induced by two-spot sonication increased 74.8% of that induced by one-spot sonication (both schemes used the same total dose of EBD). Compared to the untreated side brain, brain EBD accumulation on the treated side was 2.80-fold and 3.87-fold that on the untreated side after one-spot and two-spot sonication treatment, respectively (Figure 6). Regarding the total dose of EBD/MBs administered and the brain EBD accumulation, a two-spot sonication scheme is more efficient on brain drug delivery than a one-spot sonication scheme.

Eight mice treated with one-spot sonication and three mice treated with two-spot sonication were subjected to a histology study (Figure 7 and Table 2). For the one-spot sonication scheme, 4 out of 8 treated mice have grade 2 lesions in the acoustic field; for the two-spot sonication scheme, 1 out of 3 treated mice have grade 2 lesion in acoustic field; no grade 3 lesion was observed in our study. These findings are similar to the results reported by Hynynen et al. and Beccaria et al. [1,28] in which a grade 2 lesion was most frequently met in the treatment of acoustic pressure 0.3 to 1.4 MPa. Compared to one-spot sonication, two-spot sonication with an acoustic pressure of 0.56 MPa did not increase the grade of brain tissue damage in our observations. Kobus et al. [3] have conducted an animal study to investigate the safety of repeated focus ultrasound-induced BBBD; 15 mice were tested, and histology analysis revealed no tissue effect in 5 mice, micro-hemorrhage with or without selective neuronal necrosis in 8 mice, and extensive hemorrhage in 2 mice only, concluding that repeated BBBD by focus ultrasound can be performed with no or limited damage to the brain tissue. Several studies reported that focus ultrasound-induced BBBD would cause tissue effects including hemorrhagic change, edema, inflammation, apoptosis, alteration of cerebral blood flow, and suppression of neuronal activity [41]; these findings raise safety concerns of this treatment. However, several published clinical studies demonstrate that this treatment can be performed in human without adverse effect [42,43,44].

Although large numbers of preclinical studies demonstrate that focused ultrasound-induced BBBD is promising for treatment of brain tumor and neurodegenerative disease, its clinical use is still challenging. Biological effects of BBBD are still not well understood; adverse effects are limited but cannot be ignored. Finding the optimal paradigm, including sonication scheme, parameters, MBs size and dose, and therapeutic molecules, to have treatment efficacy and safety in the clinical setting, is still a difficult task.

## 5. Conclusions

Pulsed weakly focused ultrasound combined with MBs can open the BBB extensively with limited tissue damage. With a given acoustic pressure and PRF, the degree of BBBD increases as burst length and MBs dose increase. An EBD–albumin conjugate delivered to the brain stays longer than it does in plasma, which suggest that large therapeutic molecules delivered to the brain by focused ultrasound-induced BBBD may have a longer acting time than it does in plasma. Brain EBD accumulation amount correlates to its injection dose as well, which suggests that the amount of therapeutic molecules delivered to the brain may be predicted by its injection dose in case of BBBD. Ultrasound-induced BBBD lasts for 6 h at least, which suggests a temporal window to deliver therapeutic molecules to the brain. With a given therapeutic agent and microbubble dose, two-spot sonication may provide more efficiency than one-spot sonication with regard to its accumulation in brain. Since EBD combines with serum albumin to form a conjugate of 71–83 kDa in plasma, these results may represent brain delivery of therapeutic agents of a similar molecular scale.

## Figures and Tables

**Figure 1 pharmaceutics-14-01207-f001:**
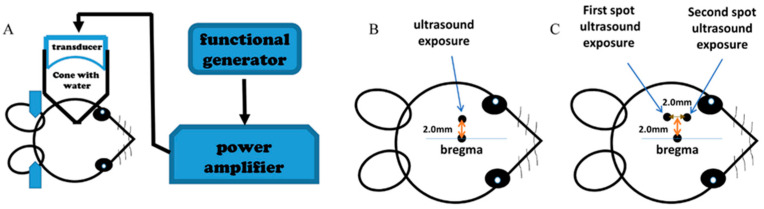
Ultrasound equipment setup and sonication locations. (**A**) Ultrasound equipment setup. (**B**) The site of one-spot sonication. (**C**) The sites of two-spot sonication.

**Figure 2 pharmaceutics-14-01207-f002:**
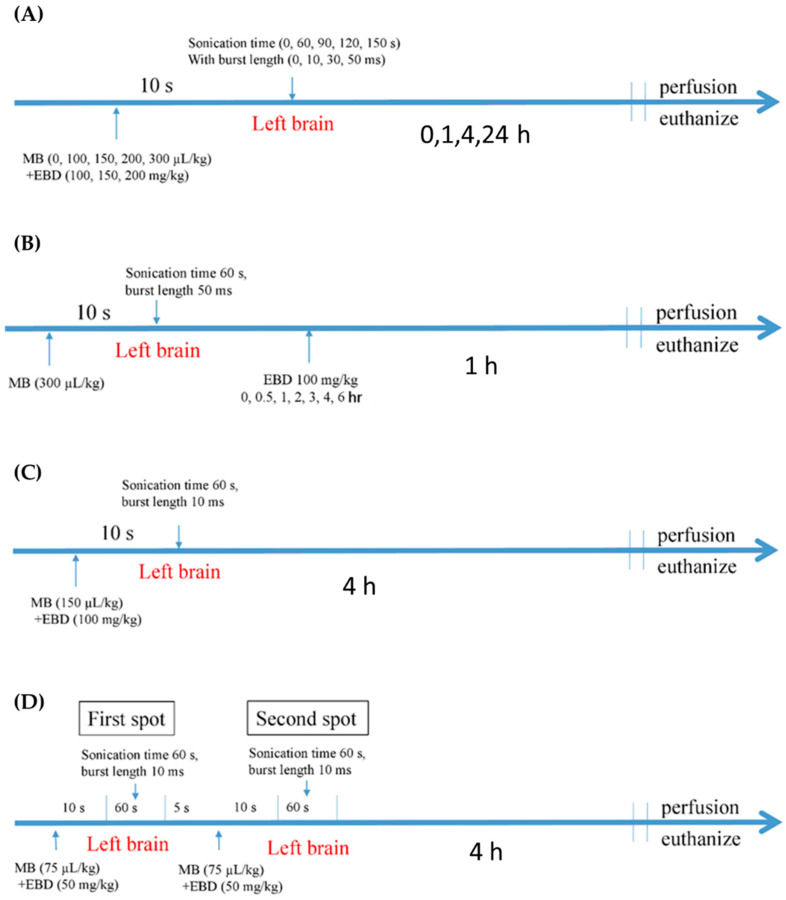
Sonication procedures. Acoustic pressure of 0.56 MPa and PRF of 1 Hz were used in all procedures. (**A**) Procedures of sonication in the parameter study; the experimental conditions are shown in Table 1. (**B**) Procedures of sonication in the BBBD dynamics study. (**C**) Procedures of one-spot sonication treatment in the sonication scheme study. (**D**) Procedures of the two-spot sonication treatment in the sonication scheme study. Each sonication used only half the dose of the EBD and MBs in Panel (**C**).

**Figure 3 pharmaceutics-14-01207-f003:**
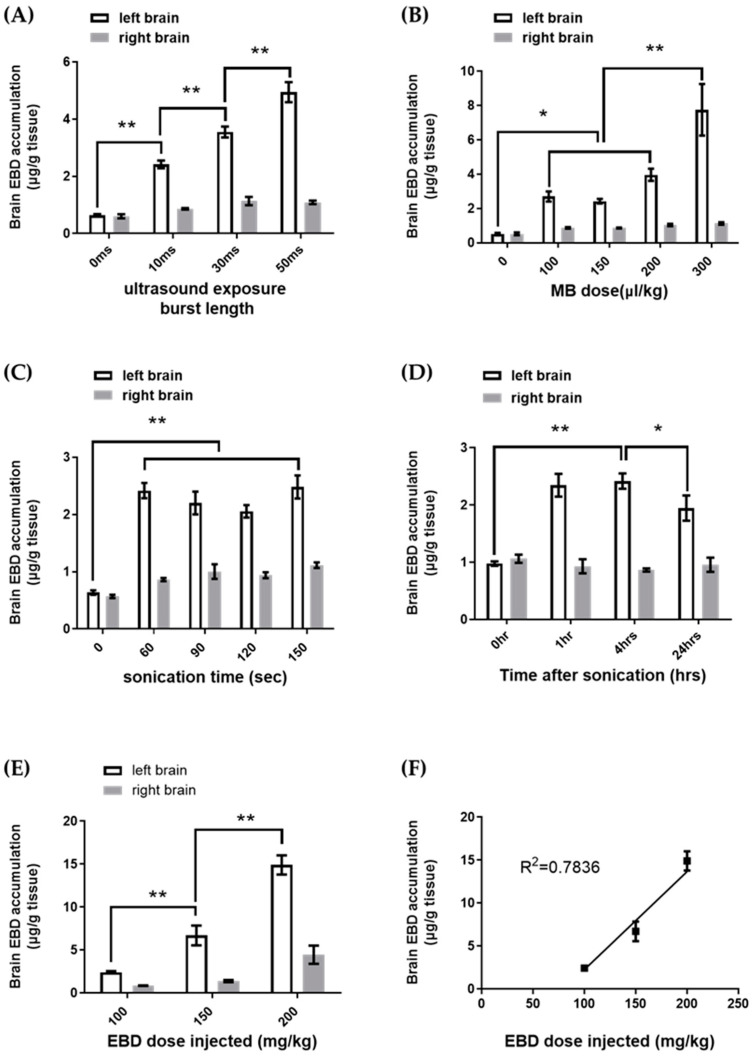
The result of the parameter study. The experimental conditions of each group are listed in Table 1. * Indicates *p* < 0.05, ** indicates *p* < 0.0001. (**A**) Brain EBD accumulation increased as burst length increased. (**B**) Brain EBD accumulation was similar with MB doses 100–200 μL/kg, but significantly increased with MB dose 300 μL/kg. (**C**) Brain EBD accumulation was similar with four different sonication time used. (**D**) Brain EBD accumulation peaked in one hour after sonication and maintained 81% of the peak level 24 h after sonication. (**E**,**F**) Brain EBD accumulation is correlated to the EBD dose injected in the case of BBBD, with a correlation coefficient of 0.7836.

**Figure 4 pharmaceutics-14-01207-f004:**
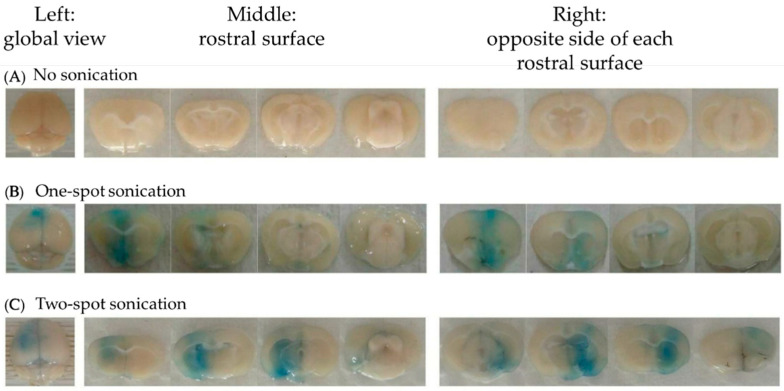
The rostral surface of four coronal sections from mice brain with BBBD induced by one-spot sonication and two-spot sonication in the sonication scheme study. Left: global view; middle: posterior side; right: anterior side. (**A**) EBD 100 mg/kg plus MB 150 μL/kg injection only, no sonication. (**B**) One-spot sonication (EBD 100 mg/kg plus MB 150 μL/kg). (**C**) Two-spot sonication (EBD 50 mg/kg plus MB 75 μL/kg for each sonication). The stained area involved all slices. The left, middle, and right slices show the EBD distributed across the whole thickness of the treated side brain.

**Figure 5 pharmaceutics-14-01207-f005:**
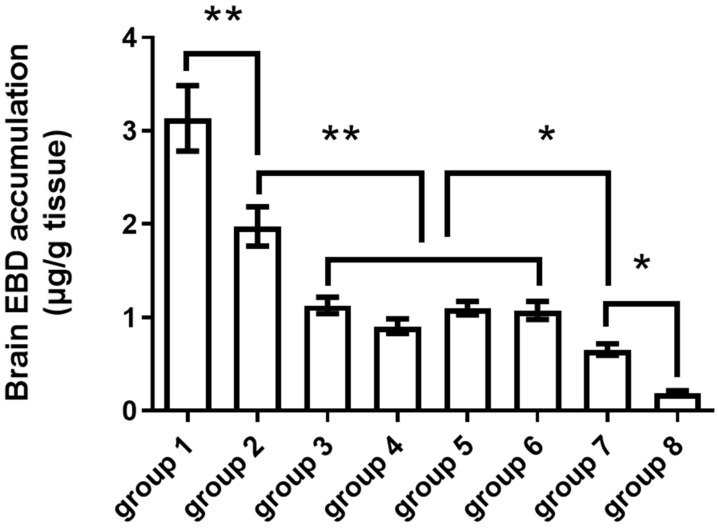
Duration of BBBD. The degree of BBBD was the greatest immediately after sonication, BBBD effect lasts for 6 h. * Indicates *p* < 0.05, ** indicates *p* < 0.01. Group 1: EBD injection after sonication 0 h (*n* = 9); Group 2: EBD injection after sonication 0.5 h (*n* = 12); Group 3: EBD injection after sonication 1 h (*n* = 9); Group 4: EBD injection after sonication 2 h (*n* = 9); Group 5: EBD injection after sonication 3 h (*n* = 9); Group 6: EBD injection after sonication 4 h (*n* = 9); Group 7: EBD injection after sonication 6 h (*n* = 3); Group 8: EBD only, no sonication (*n* = 6).

**Figure 6 pharmaceutics-14-01207-f006:**
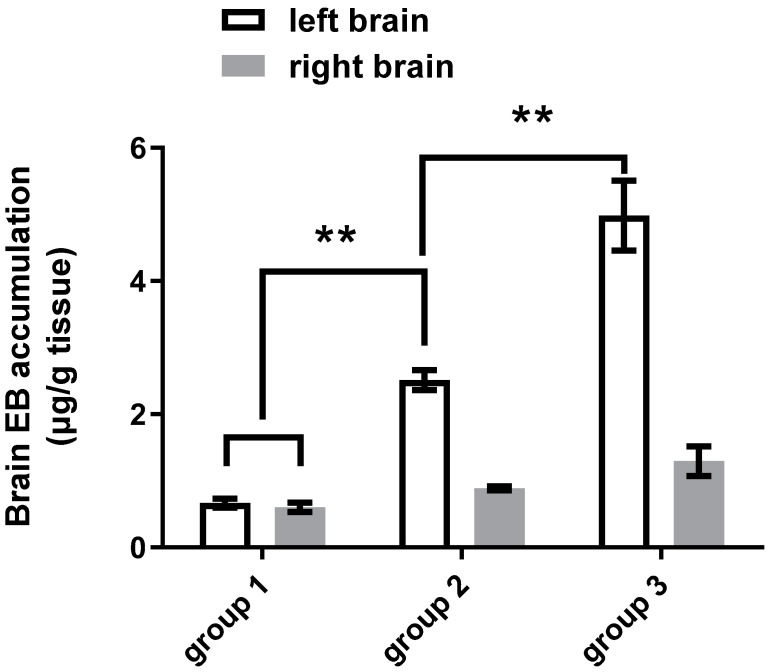
Comparison of BBBD induced by one-spot and two-spot sonication. Brain EBD accumulation of two-spot sonication-treated mice increased 74.8% compared to that of one-spot sonication-treated mice. Brain EBD accumulation on the ultrasound-treated side was 2.80-fold and 3.87-fold that of the untreated side after one-spot and two-spot sonication treatment, respectively; ** indicates *p* < 0.0001.

**Figure 7 pharmaceutics-14-01207-f007:**
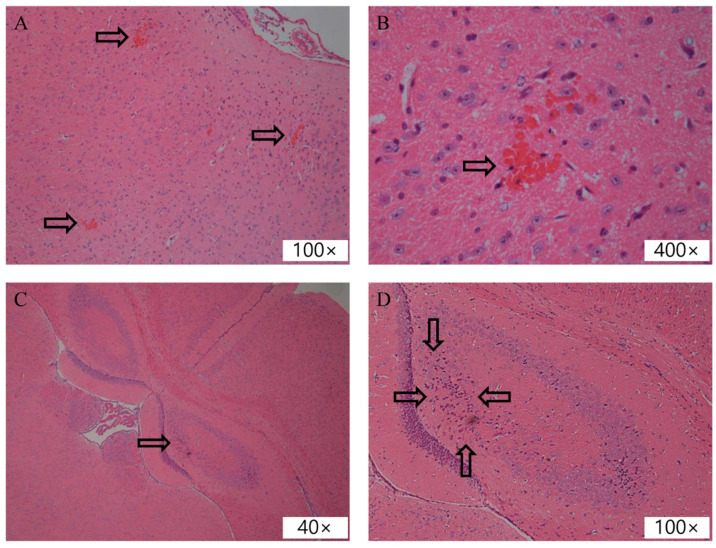
Tissue sections of brain parenchyma with hematoxylin and eosin stain. (**A**) Grade 2 lesion; the arrows indicate scattered microhemorrhages. This mouse was treated with one-spot sonication. (**B**) Magnified view of the hemorrhage in Panel (**A**); (**C**) Grade 2 lesion; the arrow indicates degenerated neurons. This mouse was treated with two-spot sonication. (**D**) Magnified view of Panel (**C**).

**Table 1 pharmaceutics-14-01207-t001:** Experimental conditions of the parameter study. An acoustic pressure of 0.56 MPa and PRF of 1 Hz were used in all experimental groups.

	Parameter	Acoustic Pressure and PRF	Sonication Burst Length (ms)	Microbubble Dose (μL/kg)	Sonication Time (s)	Euthanization Time after Sonication (h)	EBD dose Injected (mg/kg)	n
Group								
1		Acoustic pressure 0.56 MPa, PRF 1 Hz	**0**	150	-	4	100	9
			**10**		60			15
			**30**					5
			**50**					5
2			10	**0**	60	4	100	4
				**100**				4
				**150**				15
				**200**				4
				**300**				4
3			-	150	**0**	4	100	9
			10		**60**			15
					**90**			3
					**120**			5
					**150**			5
4			10	150	60	**0**	100	3
						**1**		3
						**4**		15
						**24**		3
5			10	150	60	4	**100**	15
							**150**	3
							**200**	3

**Table 2 pharmaceutics-14-01207-t002:** Histology study for brain tissue damage evaluation.

Histology Study	One-Spot Sonication	Two-Spot Sonication
Acoustic parameter	Acoustic pressure 0.56 MPa, PRF 1 Hz, burst length 10 ms, sonication time 60 s, MBs 150 μL/kg	Acoustic pressure 0.56 MPa, PRF 1 Hz, burst length 10 ms, sonication time 60 s, MBs 75 μL/kg for each sonication
n	8	3
Grade 0 lesion	0	0
Grade 1 lesion	4	2
Grade 2 lesion	4	1
Grade 3 lesion	0	0

## Data Availability

Not applicable.

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
