# Peer review of "Influence of Acoustic Parameters and Sonication Schemes on Transcranial Blood–Brain Barrier Disruption Induced by Pulsed Weakly Focused Ultrasound"

_pharmaceutics, 2022, doi:10.3390/pharmaceutics14061207_

Round 1

Reviewer 1 Report

The authors present an interesting experimental study of the different factors that allow a better disruption of the transcranial blood-brain barrier by weakly focused pulsed ultrasound, studying the influence of factors such as burst length, microbubble dose and number of spot sonication sites.

However, a little more discussion of potential risks or side effects that may be associated with the disruption of BBB for hours is lacking.  Although histologic sections are discussed for a qualitative assessment of brain damage, the manuscript should include in the introduction some more information explaining the previous literature on which they rely to assume that weakly diffused disruption is safe.

On the other hand, I find that the bibliography does not include some recent publications on the subject. There are very recent articles on BBB disruption by ultrasound, some of them even published in 2022, which should be included in the manuscript as a starting point to introduce the state of the art in this field.

Throughout the results, a great disparity of sample sizes is observed for the different conditions studied, which could call into question the validity of the conclusions drawn from the statistical analysis. This critical point should be justified and discussed by the authors.

In the discussion, the first three paragraphs contain a lot of information that does not focus on the discussion of the results of the work. Although some sentences to contextualize the results might be appropriate, some of the information included here would belong more in the introduction section.

Minor changes:

The location of figures 2 and 3 and their legends must have suffered some layout problem when they were inserted, which makes them difficult to understand. For example, line 272 is not clear from which other line of the manuscript it comes from.

The legend of figure 4 should explain the different graphs included in it and what the letters A-E correspond to. A figure and its legend are expected to be self-explanatory.

Author Response

The authors present an interesting experimental study of the different factors that allow a better disruption of the transcranial blood-brain barrier by weakly focused pulsed ultrasound, studying the influence of factors such as burst length, microbubble dose and number of spot sonication sites.

However, a little more discussion of potential risks or side effects that may be associated with the disruption of BBB for hours is lacking.  Although histologic sections are discussed for a qualitative assessment of brain damage, the manuscript should include in the introduction some more information explaining the previous literature on which they rely to assume that weakly diffused disruption is safe.

Response:

We add this paragraph in discussion (line 390-400):

  1. Kobus et al. [3] have conducted an animal study to investigate safety of repeated focus ultrasound induced BBBD, 15 mice were tested, histopathology analysis revealed that no tissue effect in 5 mice, micro-hemorrhage with or without selective neuronal necrosis in 8 mice, extensive hemorrhage in 2 mice only; they concluded that repeated BBBD by focus ultrasound can be performed with no or limited damage to the brain tissue. Several studies reported that focus ultrasound induced BBBD would cause tissue effects including hemorrhagic change, edema, inflammation, apoptosis, alteration of cerebral blood flow, and suppression of neuronal activity [40]; these findings raise safety concerns of this treatment, however several published clinical studies demonstrate that this treatment can be performed in human without adverse effect. [41-43]

On the other hand, I find that the bibliography does not include some recent publications on the subject. There are very recent articles on BBB disruption by ultrasound, some of them even published in 2022, which should be included in the manuscript as a starting point to introduce the state of the art in this field.

Response:

 We add 8 references which are published between 2018 to 2022.

  1. Abrahao, A., Meng, Y., Llinas, M., Huang, Y., Hamani, C., Mainprize, T., . . . Zinman, L. (2019). First-in-human trial of blood-brain barrier opening in amyotrophic lateral sclerosis using MR-guided focused ultrasound. Nat Commun, 10(1), 4373. doi:10.1038/s41467-019-12426-9
  2. Gandhi, K., Barzegar-Fallah, A., Banstola, A., Rizwan, S. B., & Reynolds, J. N. J. (2022). Ultrasound-Mediated Blood-Brain Barrier Disruption for Drug Delivery: A Systematic Review of Protocols, Efficacy, and Safety Outcomes from Preclinical and Clinical Studies. Pharmaceutics, 14(4). doi:10.3390/pharmaceutics14040833
  3. Lipsman, N., Meng, Y., Bethune, A. J., Huang, Y., Lam, B., Masellis, M., . . . Black, S. E. (2018). Blood-brain barrier opening in Alzheimer's disease using MR-guided focused ultrasound. Nat Commun, 9(1), 2336. doi:10.1038/s41467-018-04529-6
  4. Mainprize, T., Lipsman, N., Huang, Y., Meng, Y., Bethune, A., Ironside, S., . . . Hynynen, K. (2019). Blood-Brain Barrier Opening in Primary Brain Tumors with Non-invasive MR-Guided Focused Ultrasound: A Clinical Safety and Feasibility Study. Sci Rep, 9(1), 321. doi:10.1038/s41598-018-36340-0
  5. Todd, N., Angolano, C., Ferran, C., Devor, A., Borsook, D., & McDannold, N. (2020). Secondary effects on brain physiology caused by focused ultrasound-mediated disruption of the blood-brain barrier. J Control Release, 324, 450-459. doi:10.1016/j.jconrel.2020.05.040
  6. Walsh, A. P. G., Gordon, H. N., Peter, K., & Wang, X. (2021). Ultrasonic particles: An approach for targeted gene delivery. Adv Drug Deliv Rev, 179, 113998. doi:10.1016/j.addr.2021.113998
  7. Wu, S. K., Tsai, C. L., Huang, Y., & Hynynen, K. (2020). Focused Ultrasound and Microbubbles-Mediated Drug Delivery to Brain Tumor. Pharmaceutics, 13(1). doi:10.3390/pharmaceutics13010015
  8. Zhang, M., Fabiilli, M., Carson, P., Padilla, F., Swanson, S., Kripfgans, O., & Fowlkes, B. (2010). Acoustic Droplet Vaporization for the Enhancement of Ultrasound Thermal Therapy. Proc IEEE Ultrason Symp, 2010, 221-224. doi:10.1109/ULTSYM.2010.0054

Throughout the results, a great disparity of sample sizes is observed for the different conditions studied, which could call into question the validity of the conclusions drawn from the statistical analysis. This critical point should be justified and discussed by the authors.

Response: Thanks for reviewer’s guidance. Because of limited budget, we cannot afford more larger sample size. The key parameter set is burst length 10ms, microbubble dose 150uL/kg, sonication time 60sec, Evan blue dose 100mg/kg, euthanization time 4 hours after sonication. This parameter set is used in each group of this study for comparison and is also frequently used in other published animal studies. We used a larger sample size of this key parameter set (n=15) to confirm accuracy of this benchmark data. Although sample disparity exists, the accuracy of benchmark data for comparison is very reliable.

In the discussion, the first three paragraphs contain a lot of information that does not focus on the discussion of the results of the work. Although some sentences to contextualize the results might be appropriate, some of the information included here would belong more in the introduction section.

Response:

We move the first paragraph to introduction section (line 84-94) in order to make discussion section more focused.

Minor changes:

The location of figures 2 and 3 and their legends must have suffered some layout problem when they were inserted, which makes them difficult to understand. For example, line 272 is not clear from which other line of the manuscript it comes from.

Response:

We revise the figure 3 description on line 244-251 and re-layout the manuscript.

The legend of figure 4 should explain the different graphs included in it and what the letters A-E correspond to. A figure and its legend are expected to be self-explanatory.

Response:

We make some modifications of figure 4 legend to make reading more easily.

Reviewer 2 Report

The manuscript Influence of acoustic parameters and sonication schemes on transcranial blood-brain barrier disruption induced by pulsed weakly focused ultrasound by Yu-Hone Hsu et al. describes the in vivo study on the optimization of the blood-brain barrier permeability enhancement by sonication. The subject is important, as this phenomenon was not yet been studied in detail and improvement of drug delivery to the brain would grant a possibility to improve current treatment methods for brain diseases. The submitted article is written rather good and fits the scope of the journal Pharmaceutics.

However, there are some issues that should be addressed to improve the quality of the manuscript:

  1. Were there any exclusion criteria for the experiments? Those are not stated.
  2. What was the method for choosing the amount of specimens for each group? I understand that there can be a group of 9 or 15 mice as some experiments duplicate but some groups contain 3 or 5 specimens. This way to write that “33% of the treated mice have grade 2 lesion in acoustic field” (line 401) seems inappropriate. Add an additional description, i.e. “1 out of 3”. This also applies to the sentence “compared to one spot sonication…”(lines 404-406).
  3. How long did the actual disruption last? You mention that BBB was closed after 6h (line 360) to state that “At 6 hours after sonication, brain EBD delivery was still significant”(lines 380-381). Please rewrite to reflect on the findings. Based on the data it is impossible to assess whether the BBBD was reversible, this is only a speculation.
  4. The study should contain the limitations and drawbacks of such a treatment in the discussion section.
  5. All the graphics and tables from subchapter 3.2 should be put next to their corresponding citations in-text. Currently it is hard to follow.
  6. Please fix the citations. They do not appear in order of appearance in text, as the first citation in line 60 is [5,10,19,24,26,29,31,38]. Also, there is one citation in Harvard style (line 187).
  7. Caption of Figure 2 is between Figures 1 and 3 on page 7 while the figure itself is on the 8th page, after Figure 3.
  8. Please combine the same affiliations (2-6).
  9. Delete the fragment in lines 193-195.

Author Response

The manuscript Influence of acoustic parameters and sonication schemes on transcranial blood-brain barrier disruption induced by pulsed weakly focused ultrasound by Yu-Hone Hsu et al. describes the in vivo study on the optimization of the blood-brain barrier permeability enhancement by sonication. The subject is important, as this phenomenon was not yet been studied in detail and improvement of drug delivery to the brain would grant a possibility to improve current treatment methods for brain diseases. The submitted article is written rather good and fits the scope of the journal Pharmaceutics.

However, there are some issues that should be addressed to improve the quality of the manuscript:

  1. Were there any exclusion criteria for the experiments? Those are not stated.

Response:

Thanks for reviewer’s guidance. We exclude some data which is due to eventful sonication operation, such as cone water not degassed well, or delayed timing of Evan blue injection. We enrolled all data which is obtained from uneventful sonication operation to avoid investigator’s subjective opinion on the data. Fortunately we have not marked outliers.

  1. What was the method for choosing the amount of specimens for each group? I understand that there can be a group of 9 or 15 mice as some experiments duplicate but some groups contain 3 or 5 specimens. This way to write that “33% of the treated mice have grade 2 lesion in acoustic field” (line 401) seems inappropriate. Add an additional description, i.e. “1 out of 3”. This also applies to the sentence “compared to one spot sonication…”(lines 404-406).

Response:

In later stage of the study, our mice supply is not as sufficient as it was in early stage because of limited budget. The key parameter set is burst length 10ms, microbubble dose 150uL/kg, sonication time 60sec, Evan blue dose 100mg/kg, euthanization time 4 hours after sonication. This parameter set is used in each group of this study for comparison, and is also frequently used in other published animal studies. We used a larger sample size of this key parameter set (n=15) to confirm accuracy of this benchmark data. The inappropriate descriptions were corrected according to reviewer’s guidance.

  1. How long did the actual disruption last? You mention that BBB was closed after 6h (line 360) to state that “At 6 hours after sonication, brain EBD delivery was still significant” (lines 380-381). Please rewrite to reflect on the findings. Based on the data it is impossible to assess whether the BBBD was reversible, this is only a speculation.

Response: Thanks to reviewer for pointing out our error.

Indeed, our data cannot prove that focused ultrasound induced BBB disruption is reversible. We delete the sentence “though BBB closed 6 hours after sonication in our study”. Reversibility of BBB disruption is demonstrated by other published studies (reference 2,37,38,39).

  1. The study should contain the limitations and drawbacks of such a treatment in the discussion section.

Response: As reviewer’s said, clinical use of this treatment is still difficult in real world. We add a paragraph in discussion:
“Although large numbers of preclinical studies demonstrate that focused ultrasound induced BBBD is promising for treatment of brain tumor and neurodegenerative disease, its clinical use is still challenging. Biological effects of BBBD are still not well understood, adverse effects are limited but cannot be ignored. Finding of optimal paradigm, including sonication scheme, parameters, MBs size and dose, and therapeutic molecules to have treatment efficacy and safety in clinical setting is still a difficult task.”

  1. All the graphics and tables from subchapter 3.2 should be put next to their corresponding citations in-text. Currently it is hard to follow.

Response: We have revised the figures and re-layout the manuscript

  1. Please fix the citations. They do not appear in order of appearance in text, as the first citation in line 60 is [5,10,19,24,26,29,31,38]. Also, there is one citation in Harvard style (line 187).

Response: We have fixed the citations and re-layout the reference.

  1. Please combine the same affiliations (2-6).

Response: We have combined the same affiliations (2-6) and re-layout the manuscript

  1. Delete the fragment in lines 193-195.

Response: We have deleted the fragment in lines.

Reviewer 3 Report

The submitted article for review corresponds to the subject of the authoritative scientific journal Pharmaceutics (ISSN: 1999-4923)

Influence of acoustic parameters and sonication schemes on transcranial blood-brain barrier disruption induced by pulsed weakly focused ultrasound

The title is good and quite specific and attracts the reader's interest so that it is easy to understand.

The topic of scientific research is quite interesting and relevant.

The scientific article is logically built, corresponds to the principles of presenting scientific information and research.

I think that the article used enough tables, graphics and illustrations.

I note that the authors of the article treated the analysis of the problem quite thoroughly and used the necessary number of sources of information to prepare the submitted manuscript.

Author's notes

Abstract

The abstract looks very general. The authors should briefly mention the importance of the research work. Add more discussions related to the study, the value of the study results, and future research directions. It is surprising that there are 16 citations per sentence. We believe that it is necessary to completely revise the list of cited sources of information (References). There are no sources dated from 2017. The authors are encouraged to seriously consider this issue.

Add a few more keywords.

  1. Results and Discussion

Line 276, 286, 295

Figure 4, 5, 6  -  is it possible for the authors to present a clearer image?

In these figures, the color is unsuccessfully chosen, there is no visual understanding (perception) of the established measurement error.

  1. Materials and Methods

Please add - indicate the date (year) of the beginning of the research and the date (year) of the end of this study.

  1. Conclusions

I recommend the authors to revise the text of this section of the scientific article. Authors need to clearly state the results of this scientific study and present its results in more detail.

Funding

Perhaps the authors missed the full names of the sources of funding - They did not indicate the country.

References

I was unable to validate references #4 (line 442), #18 (line 479), #28 (line 505), #32 (line 512), #33 (line 514), #39 (line 528).

I recommend for publication the scientific article "Influence of acoustic parameters and sonication schemes on transcranial blood-brain barrier disruption induced by pulsed weakly focused ultrasound" - pharmaceutics-1726914 after the comments have been corrected. Deep processing of sections of a scientific article: Abstract, References.

Author Response

The submitted article for review corresponds to the subject of the authoritative scientific journal Pharmaceutics (ISSN: 1999-4923)

Influence of acoustic parameters and sonication schemes on transcranial blood-brain barrier disruption induced by pulsed weakly focused ultrasound

The title is good and quite specific and attracts the reader's interest so that it is easy to understand.The topic of scientific research is quite interesting and relevant.The scientific article is logically built, corresponds to the principles of presenting scientific information and research.I think that the article used enough tables, graphics and illustrations.

I note that the authors of the article treated the analysis of the problem quite thoroughly and used the necessary number of sources of information to prepare the submitted manuscript.

Author's notes

Abstract

The abstract looks very general. The authors should briefly mention the importance of the research work. Add more discussions related to the study, the value of the study results, and future research directions.

Response: We have revised the abstract body as following:

Abstract: Pulsed ultrasound combined with microbubbles use can disrupt blood-brain barrier (BBB) temporarily, this technique opens a temporal window to deliver large therapeutic molecules into brain tissue. There are published studies to discuss the efficacy and safety of different ultrasound parameters, microbubble dose and size, and sonication scheme on BBB disruption, but optimal paradigm is still under investigation. Our study is aimed to investigate how different sonication parameters and time, and microbubble dose affect BBB disruption; dynamics of BBB disruption; and efficacy of different sonication scheme on BBB disruption.

Method: We used pulsed weakly focused ultrasound to open BBB of C57/B6 mice. Evans blue dye (EBD) was used to determine the degree of BBB disruption. With a given acoustic pressure 0.56MPa and pulse repetitive frequency 1Hz, burst length 10ms to 50ms, microbubble 100μl/kg to 300μl/kg and sonication time 60s to 150s were used to open BBB for parameter study; brain EBD accumulation was measured at 1, 4, 24 hours after sonication for time-response relationship study; EBD 100mg/kg to 200mg/kg was administered for dose-response relationship study; EBD injection 0 to 6 hours after sonication was performed for BBB disruption dynamic study; brain EBD accumulation induced by one sonication and two sonications was investigated to study the effectiveness on BBB disruption; histology study was performed for brain tissue damage evaluation.

Results: Pulsed weakly focused ultrasound opens BBB extensively. Longer burst length and larger microbubble dose result in more degree of BBB disruption; sonication time longer than 60s did not increase BBB disruption; brain EBD accumulation peaks 1 hour after sonication and remains 81% of peak level 24 hours after sonication; EBD dose administered correlates to brain EBD accumulation; BBB disruption decreases as time goes on after sonication and lasts for 6 hours at least; brain EBD accumulation induced by two sonication increases 74.8% of that induced by one sonication. There was limited adverse effect associated with sonication including petechial hemorrhages and mild neuronal degeneration.

Conclusion: BBB can be opened extensively and reversibly by pulsed weakly focused ultrasound with limited brain tissue damage. Since EBD combines with albumin in plasma to form a conjugate of 83kDa, these results may simulate ultrasound induced brain delivery of therapeutic molecules of this size scale. The result of our study may contribute to task of finding the optimal paradigm of focused ultrasound induced BBB disruption.

It is surprising that there are 16 citations per sentence. We believe that it is necessary to completely revise the list of cited sources of information (References). There are no sources dated from 2017. The authors are encouraged to seriously consider this issue.

Add a few more keywords.

 Response: The sentence containing too many citations have been revised. We also add 8 new references published between 2018 and 2022, and we add two key words (Neurodegenerative diseases, Microbubbles)

The added references are listed below:

  1. Abrahao, A., Meng, Y., Llinas, M., Huang, Y., Hamani, C., Mainprize, T., . . . Zinman, L. (2019). First-in-human trial of blood-brain barrier opening in amyotrophic lateral sclerosis using MR-guided focused ultrasound. Nat Commun, 10(1), 4373. doi:10.1038/s41467-019-12426-9
  2. Gandhi, K., Barzegar-Fallah, A., Banstola, A., Rizwan, S. B., & Reynolds, J. N. J. (2022). Ultrasound-Mediated Blood-Brain Barrier Disruption for Drug Delivery: A Systematic Review of Protocols, Efficacy, and Safety Outcomes from Preclinical and Clinical Studies. Pharmaceutics, 14(4). doi:10.3390/pharmaceutics14040833
  3. Lipsman, N., Meng, Y., Bethune, A. J., Huang, Y., Lam, B., Masellis, M., . . . Black, S. E. (2018). Blood-brain barrier opening in Alzheimer's disease using MR-guided focused ultrasound. Nat Commun, 9(1), 2336. doi:10.1038/s41467-018-04529-6
  4. Mainprize, T., Lipsman, N., Huang, Y., Meng, Y., Bethune, A., Ironside, S., . . . Hynynen, K. (2019). Blood-Brain Barrier Opening in Primary Brain Tumors with Non-invasive MR-Guided Focused Ultrasound: A Clinical Safety and Feasibility Study. Sci Rep, 9(1), 321. doi:10.1038/s41598-018-36340-0
  5. Todd, N., Angolano, C., Ferran, C., Devor, A., Borsook, D., & McDannold, N. (2020). Secondary effects on brain physiology caused by focused ultrasound-mediated disruption of the blood-brain barrier. J Control Release, 324, 450-459. doi:10.1016/j.jconrel.2020.05.040
  6. Walsh, A. P. G., Gordon, H. N., Peter, K., & Wang, X. (2021). Ultrasonic particles: An approach for targeted gene delivery. Adv Drug Deliv Rev, 179, 113998. doi:10.1016/j.addr.2021.113998
  7. Wu, S. K., Tsai, C. L., Huang, Y., & Hynynen, K. (2020). Focused Ultrasound and Microbubbles-Mediated Drug Delivery to Brain Tumor. Pharmaceutics, 13(1). doi:10.3390/pharmaceutics13010015
  8. Zhang, M., Fabiilli, M., Carson, P., Padilla, F., Swanson, S., Kripfgans, O., & Fowlkes, B. (2010). Acoustic Droplet Vaporization for the Enhancement of Ultrasound Thermal Therapy. Proc IEEE Ultrason Symp, 2010, 221-224. doi:10.1109/ULTSYM.2010.0054

Line 276, 286, 295

Figure 4, 5, 6 -  is it possible for the authors to present a clearer image?

In these figures, the color is unsuccessfully chosen, there is no visual understanding (perception) of the established measurement error.

 Response: Thanks for reviewer’s guidance. We have revised the display of figure 4, 5, 6.

Materials and Methods

Please add - indicate the date (year) of the beginning of the research and the date (year) of the end of this study.

 Response: The study is conducted from 1st August 2016 to 31th July, 2019. We have added it to the text.

Conclusions

I recommend the authors to revise the text of this section of the scientific article. Authors need to clearly state the results of this scientific study and present its results in more detail.

 Response: Thanks for reviewer’s guidance, we have re-written Conclusions as following:

Pulsed weakly focused ultrasound combined with MBs can open BBB extensively with limited tissue damage. With a given acoustic pressure and PRF, the degree of BBBD increases as burst length and MBs dose increase. EBD-albumin conjugate delivered to brain stays more time than it does in plasma, which suggest that large therapeutic molecules delivered to brain by focused ultrasound induced BBBD may have a longer acting time than it does in plasma. Brain EBD accumulation amount correlates to its injection dose well, which suggests that the amount of therapeutic molecules delivered to brain may be predicted by its injection dose in case of BBBD. Ultrasound induced BBBD lasts for 6 hours at least, which suggests a temporal window to deliver therapeutic molecules to the brain. With a given therapeutic agent and microbubble dose, two-spot sonication may provide more efficiency than one-spot sonication in regard to its accumulation in brain. Since EBD combines with serum albumin to form a conjugate of 71-83kDa in plasma, these results may represent brain delivery of therapeutic agents of similar molecular scale.

Funding

Perhaps the authors missed the full names of the sources of funding - They did not indicate the country.

 Response: We have added the country of the sources of funding.

Funding:

This study was supported by Ministry of Health and Welfare (Taiwan) (MOHW111-TDU-B-221-114017 to T.-T.W.); Ministry of Science and Technology (Taiwan) (MOST 108-2314-B-038-061-MY3 to T.-T.W.); the Featured Areas Research Center Program within the framework of the Higher Education Sprout Project by the Ministry of Education (Taiwan) (MOE) (DP2-110-21121-03-C-02-01 and DP2-110-21121-03-C-02-02). Kaohsiung Veterans General Hospital (Taiwan) (VGHKS109-102)

References

I was unable to validate references #4 (line 442), #18 (line 479), #28 (line 505), #32 (line 512), #33 (line 514), #39 (line 528).

Response: We had removed references #4, #18, #28, #33, #39. In regard to #32, Rawson RA. 1943. The binding of T-1824 and structurally related diazo dyes by the plasma proteins. Amer. J. Physiol 137: 708-717, the doi is https://doi.org/10.1152/ajplegacy.1943.138.5.708 , the full text can be found in SCI-HUB. The T-1824 dye is Evans blue.

I recommend for publication the scientific article "Influence of acoustic parameters and sonication schemes on transcranial blood-brain barrier disruption induced by pulsed weakly focused ultrasound" - pharmaceutics-1726914 after the comments have been corrected. Deep processing of sections of a scientific article: Abstract, References.

Submission Date

29 April 2022

Date of this review

16 May 2022 21:27:39

Round 2

Reviewer 1 Report

In line 191 there is a typographical error ("histopathology"). However, in the corresponding results and discussion sections, histology is used instead of histopathology. The terms used throughout the manuscript should be clarified and correspond to those used in the methodology section.

Author Response

Comments and Suggestions for Authors

In line 191 there is a typographical error ("histopathology"). However, in the corresponding results and discussion sections, histology is used instead of histopathology. The terms used throughout the manuscript should be clarified and correspond to those used in the methodology section.

Response: Thanks for reviewer's guidance. We have corrected "histopathology" to "histology".
